# Accumulation of health complaints is associated with persistent musculoskeletal pain two years later in adolescents: The Fit Futures study

Kaja Smedbråten[1]*, Margreth Grotle[2,3], Henriette Jahre[1,2], Kåre Rønn Richardsen[1,2], Pierre Côté[4], Ólöf Anna Steingrímsdóttir[5,6], Kjersti Storheim[1,3], Christopher Sivert Nielsen[7,8], Britt Elin Øiestad[1,2]

1 Department of Rehabilitation Science and Health Technology, Oslo Metropolitan University, Oslo, Norway, 2 Department of Rehabilitation Science and Health Technology, Centre for Intelligent Musculoskeletal Health, Oslo Metropolitan University, Oslo, Norway, 3 Division of Clinical Neuroscience, Research and Communication Unit for Musculoskeletal Health (FORMI), Oslo University Hospital, Oslo, Norway, 4 Faculty of Health Sciences and Institute for Disability and Rehabilitation Research, Ontario Tech University, Oshawa, Canada, 5 Division of Mental and Physical Health, Department of Physical Health and Ageing, Norwegian Institute of Public Health, Oslo, Norway, 6 Division of Emergencies and Critical Care, Department of Research and Development, Oslo University Hospital, Oslo, Norway, 7 Division of Mental and Physical Health, Department of Chronic Diseases, Norwegian Institute of Public Health, Oslo, Norway, 8 Division of Emergencies and Critical Care, Department of Pain Management and Research, Oslo University Hospital, Oslo, Norway

* kasm@oslomet.no

**Data Availability Statement:** The data underlying the results presented in the study are available from the Fit Futures Study, but restrictions apply to

## Abstract

There is limited knowledge on the association between different health complaints and the development of persistent musculoskeletal pain in adolescents. The aims of this study were to assess whether specific health complaints, and an accumulation of health complaints, in the first year of upper-secondary school, were associated with persistent musculoskeletal pain 2 years later. We used data from a population-based cohort study (the Fit Futures Study in Norway), including 551 adolescents without persistent musculoskeletal pain at baseline. The outcome was persistent musculoskeletal pain (≥3 months) 2 years after inclusion. The following self-reported health complaints were investigated as individual exposures at baseline: asthma, allergic rhinitis, atopic eczema, headache, abdominal pain and psychological distress. We also investigated the association between the accumulated number of self-reported health complaints and persistent musculoskeletal pain 2 years later. Logistic regression analyses estimated adjusted odds ratios (ORs) with 95% confidence intervals (CIs). At the 2-year follow-up, 13.8% (95% CI [11.2–16.9]) reported persistent musculoskeletal pain 2 years later (OR 2.33, 95% CI [1.29–4.19], p = 0.01). Our analyses showed no statistically significant associations between asthma, allergic rhinitis, atopic eczema, headache or psychological distress and persistent musculoskeletal pain at the 2-year follow-up. For the accumulated number of health complaints, a higher odds of persistent musculoskeletal pain at the 2-year follow-up was observed for each additional health complaint at baseline (OR 1.33, 95% CI [1.07–1.66], p = 0.01). Health care providers might need to take preventive

the availability of these data, which were used under license for the current study, and so are not publicly available. Data can be made available from the Fit Futures Study upon application (fitfutures@uit.no).

**Funding:** This study was supported by the Norwegian Fund for Post-Graduate Training in Physiotherapy (grant number 105597) to KS. The funders had no role in study design, data collection and analysis, decision to publish, or preparation of the manuscript.

**Competing interests:** The authors have declared that no competing interests exist.

actions in adolescents with abdominal pain and in adolescents with an accumulation of health complaints to prevent development of persistent musculoskeletal pain. The potential multimorbidity perspective of adolescent musculoskeletal pain is an important topic for future research to understand the underlying patterns of persistent pain conditions in adolescents.

## Introduction

Musculoskeletal pain is common already in young ages, with a prevalence of up to 40% among children and adolescents [1]. Adolescents with persistent musculoskeletal pain report difficulties in daily activities [2] and reduced health-related quality of life [3]. Furthermore, experiencing musculoskeletal pain in adolescence is a predictor of future pain [4], and of receiving sickness benefits [5] and long-term social welfare benefits [6] in adulthood. Given these consequences, it is important to draw attention to risk factors of persistent musculoskeletal pain onset in adolescence to be able to develop appropriate preventive strategies.

Importantly, also headache [1, 7], abdominal pain [1] and mental health problems [8] are common health complaints in children and adolescents. Furthermore, medical conditions, such as asthma, eczema [9, 10] and allergic rhinitis [9] are frequent. Several of these health complaints commonly co-occur with musculoskeletal pain [11–13], however, whether the presence of different health complaints increase the risk of developing musculoskeletal pain is unclear. A systematic review suggested that negative emotional symptoms might increase the risk of developing musculoskeletal pain in children and adolescents [14]. Studies have also indicated an association between abdominal pain [15] and headache [15–18] with later musculoskeletal pain. For the association between other health issues, including medical conditions, and musculoskeletal pain onset in children and adolescents, the existing findings are inconsistent [14].

A cross-sectional study by Dominick et al [19] demonstrated that several discrete chronic physical conditions, and the number of conditions, were independently associated with chronic pain in the general population aged 15 years and over. Furthermore, a cross-sectional study suggested that adolescents with two additional health complaints, were more likely to report low back pain compared to those with one additional health complaint [13]. Thus, investigating both specific and an accumulation of health complaints might be of interest in relation to persistent musculoskeletal pain. According to a previous cohort study of 9-12-year-olds, an increasing number of physical and psychological symptoms increased the odds of future neck pain [20]. Furthermore, a strong association between a combination of persistent low back pain, persistent headache and asthma in adolescence and future persistent low back pain has been reported [21]. To the best of our knowledge, no previous cohort studies have investigated the association between an accumulation of health complaints in adolescence, including medical diseases, psychological and somatic symptoms, and the onset of persistent musculoskeletal pain.

Hence, the aims of this study were to assess whether specific health complaints, and an accumulation of health complaints, in adolescents without persistent musculoskeletal pain in the first year of upper-secondary school, were associated with persistent musculoskeletal pain ($\geq$ 3 months) 2 years later.

## Material and methods

### Study design and sample

This is a population-based cohort study using data from the Fit Futures study, which was conducted in the municipalities of Tromsø and Balsfjord in Northern Norway. The first wave of

the study, Fit Futures 1 (FF1), was conducted in 2010–2011 and all first-year upper-secondary school students in the two municipalities were invited to participate. Two years later (2012–2013), all FF1 participants, and all other third-year students in the same upper-secondary schools, were invited to participate in the second wave of the study, Fit Futures 2 (FF2). In our sample we have included adolescents who participated in both waves. The assessments and data collection were conducted during school hours, at a research unit at the University Hospital of North Norway, and included an electronic questionnaire, clinical examinations and a clinical interview. Further details regarding the Fit Futures study have been presented earlier [22, 23].

Of 1117 adolescents invited to participate in FF1, 1038 (92.9%) responded. Thirty-six participants were excluded from our sample because they were older than 19 years of age at baseline. Moreover, 190 participants reported persistent musculoskeletal pain at baseline and were excluded from the sample. Finally, 23 participants were excluded because they had incomplete data on musculoskeletal pain at baseline (Fig 1). Of 789 respondents to the FF1 study that met the inclusion criteria, 551 (69.8%) participated in FF2 and had complete outcome data for the present study.

All participants provided written informed consent prior to participation. For participants under the age of 16 years, written consent of one parent/guardian was also obtained. Our study protocol was approved by the Regional Committee for Medical and Health Research Ethics (2019/599/REK Nord) in Norway and approved by the Norwegian Centre for Research Data (954769). The protocol was registered at ClinicalTrials.gov (NCT05036746). The Strengthening the Reporting of Observational Studies in Epidemiology (STROBE) guidelines were followed to ensure the reporting quality of this study [24].

## Persistent musculoskeletal pain

The outcome was persistent musculoskeletal pain assessed at the 2-year follow-up with the question "Do you have persistent or recurring pain that has lasted for 3 months or more?". In addition, pain frequency and location were assessed. We defined persistent musculoskeletal pain as pain that persists or is recurring on a weekly basis for three months or more, in one or several of the following pain sites; "shoulder", "arm/elbow", "hand", "hip", "thigh/ knee/shin", "ankle/foot", "jaw/temporomandibular joint", "neck", "upper back", "lower back" and/or "chest".

Persistent musculoskeletal pain was also measured and defined as described above at baseline. The measurement of persistent musculoskeletal pain at baseline was only used to identify the sample in our cohort study.

## Health complaints

The Fit Futures Study questionnaire included a section to measure health complaints. The presence of asthma, allergic rhinitis and atopic eczema were measured by the question "Has a doctor ever told you that you have...", followed by "asthma", "hay-fever or allergic rhinitis" and "children's eczema or atopic eczema". "Hay fever or allergic rhinitis" will in the following be termed "allergic rhinitis" and "children's eczema or atopic eczema" will be termed "atopic eczema". The response alternatives were "yes", "no" and "don't know". Those who answered "yes" were categorized as having the disease of interest. For the main analyses the response alternatives "no" and "don't know" were merged.

Psychological distress, including symptoms of depression and anxiety, was measured using the Hopkins Symptom Checklist– 10 (HSCL-10) [25, 26]. HSCL-10 has been validated in Norwegian adolescents [27]. The questionnaire includes ten items, in which each was rated on a

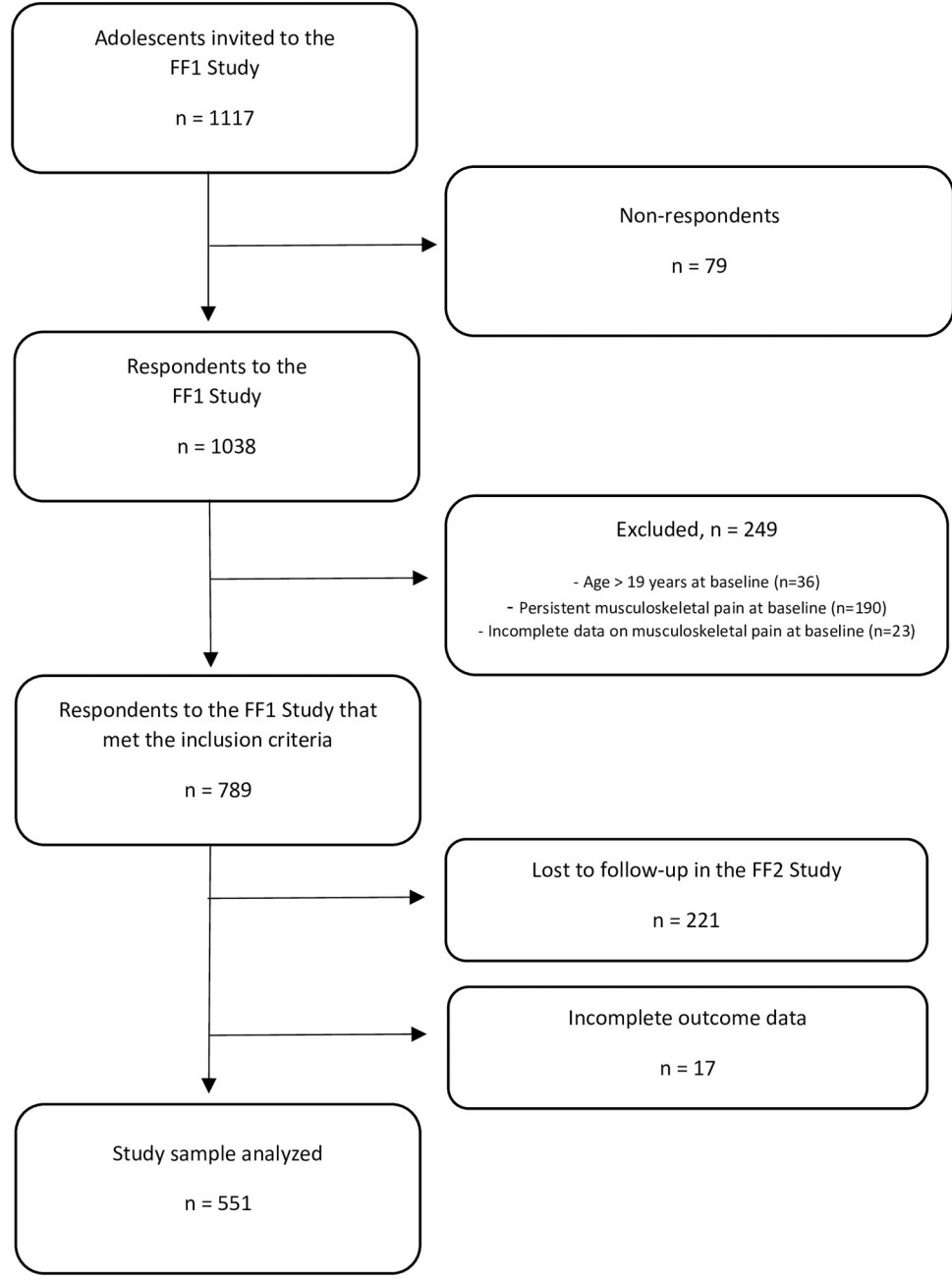

**Fig 1. Flow chart of the study sample.**

four-point scale from 1 (not at all bothered) to 4 (extremely bothered). A mean score (range 1–4) was calculated. The variable was used as a continuous variable in the analysis of the association between psychological distress and persistent musculoskeletal pain, but also as a dichotomised variable to create the variable of the accumulated number of health complaints. When dichotomised, we used a well-established cut-off score of >1.85 to define the presence of psychological distress [25].

Abdominal pain was measured by the question "During the last two months: how often did you have pain or discomfort in your stomach?" The answer options were "never", "1 to 3 times

a month", "once a week", "several times a week" and "every day". Participants who reported weekly or more frequent abdominal pain were categorised as having abdominal pain at baseline.

Headache was measured by the questions "Did you suffer from headaches the last year?" and "If you have suffered from headaches the last year; how many days per month do you suffer from headache?" The answer options were "less than 1 day", "1 to 6 days", "7 to 14 days" and "more than 14 days" per month. Participants who reported that they had suffered from headache for at least 1 to 6 days per month were categorised as having headache at baseline.

Other health complaints were measured by the question "Do you have any chronic or persistent disease?" The question was asked during a clinical interview and the reported health complaints were recoded to ICD-10 codes. The prevalence of the other health complaints, reported through this question, were too low to analyse as separate exposures, but they were included as separate health complaints when investigating the accumulated number of health complaints at baseline.

The accumulated number of health complaints was analysed as a continuous variable and included any health complaints reported at baseline (asthma, allergic rhinitis, atopic eczema, psychological distress, headache, abdominal pain and any other self-reported health complaints).

## Potential confounders

Potential confounding factors in the association between health complaints and musculoskeletal pain 2 years later were selected based on theory and previous empirical findings. Socioeconomic status [14, 28, 29] and sex [6, 30] were considered as potential confounders. In the analyses of specific health complaints, we also adjusted for all the other health complaints by generating a new variable (yes = at least one other health complaint / no = no other health complaints). Parents'employment status was used as an indicator of socioeconomic status and was measured with the questions "Is your father currently employed?" and "Is your mother currently employed?" and the answer options "full time employed", "part time employed", "unemployed", "disabled", "working at home (housewife)", "attends school / courses", "retired", "deceased", "other" and "don't know". The variables were merged and categorized as "both parents are currently employed" (fulltime or part-time), "one/neither parent is currently employed" and "don't know". Parents´ education was used as a second indicator of socioeconomic status and categorised by the highest completed education by father or mother; "at least one parent with higher education", "primary / secondary school" and "don't know". Due to a large proportion (23.8%) who did not know the educational level of their parents, parents' employment status was used as the indicator of socioeconomic status in the final models.

## Statistical analyses

Descriptive data was presented as median and range (due to skewed data) and categorical data as counts and percentages. Mann-Whitney U test (skewed data) and Chi-Square tests (categorical data) were used to compare baseline characteristics of the individuals lost to follow-up or with missing outcome data with the study sample.

Incidence rate with 95% confidence intervals (CIs) of persistent musculoskeletal pain was computed as the number of participants who reported persistent musculoskeletal pain at the 2-year follow-up divided by the total number of the study sample and reported as percentage.

Logistic regression analyses [31] were used to measure the association between baseline health complaints and the presence of persistent musculoskeletal pain at the 2-year follow-up. We analysed crude and adjusted associations with persistent musculoskeletal pain in separate

models for specific health complaints (asthma, allergic rhinitis, atopic eczema, psychological distress, headache and abdominal pain) and the accumulated number of these and other reported baseline health complaints. In the adjusted models, all potential confounders were included (sex, parents'employment status and the presence of other health complaints in the analyses of specific complaints, and sex and parents'employment status in the analysis of the accumulated number of baseline health complaints). Due to low proportion of missing values in exposures and confounders (4.0% missing cases in the adjusted models), complete case analyses were conducted. The results from the logistic regression analyses are reported as odds ratios (ORs) with 95% CIs.

Sensitivity analyses investigating the associations between asthma, allergic rhinitis, atopic eczema and persistent musculoskeletal pain at the 2-year-follow-up, excluding participants who responded "don't know" to the baseline health complaints questions, were conducted.

P-values < 0.05 were considered statistically significant. The analyses were conducted with STATA statistical software system, version 16 [32].

## Results

The study included 286 girls (51.9%) and the median age was 16 years (range 15–19) at baseline. The median number of health complaints at baseline was 1 (range 0–5). Characteristics of the study sample are presented in Table 1.

Compared to those completing the follow-up study, the group lost to follow-up or with incomplete outcome data had a lower proportion of girls, a lower proportion of adolescents with at least one parent with higher education and a higher proportion who did not know the educational level of their parents (S1 Table).

### Associations between health complaints at baseline and persistent musculoskeletal pain at 2-year follow-up

The two-year incidence of persistent musculoskeletal pain was 13.8% (95% CI [11.2–16.9]), (n = 76). Participants with baseline abdominal pain had 2.33 (95% CI [1.29–4.19], p = 0.01) higher odds of persistent musculoskeletal pain at follow-up compared to those without abdominal pain (Table 2). We found no statistically significant associations between headache, psychological distress or doctor-diagnosed asthma, allergic rhinitis or atopic eczema and persistent musculoskeletal pain at the 2-year follow-up.

Sensitivity analyses excluding individuals who answered "don't know" to the questions of doctor-diagnosed asthma, allergic rhinitis and atopic eczema, revealed comparable results as the main analyses (S2 Table).

### Association between accumulation of health complaints at baseline and persistent musculoskeletal pain at 2-year follow-up

Our analysis shows a higher odds of persistent musculoskeletal pain at the 2-year follow-up for each additional health complaint at baseline (OR 1.33, 95% CI [1.07–1.66], p = 0.01) (Table 3).

## Discussion

In this population-based cohort study of Norwegian adolescents, having abdominal pain in the first year of upper-secondary school was associated with persistent musculoskeletal pain 2 years later. Also, an accumulation of health complaints was associated with persistent musculoskeletal pain at the 2-year follow-up.

**Table 1. Baseline characteristics of the study sample.**

| Characteristics | Total |
|---|---|
| | **(n = 551)** |
| Age, y, median (min-max) | 16 (15–19) |
| Sex, girls, n (%) | 286 (51.9) |
| Parents'employment status, n (%) | |
| Both parents are currently employed | 415 (75.3) |
| One/neither parent is currently employed | 113 (20.5) |
| Don't know | 20 (3.6) |
| Missing | 3 (0.5) |
| Parents'education, n (%) | |
| At least one parent with higher education | 281 (51.0) |
| Primary / secondary school | 126 (22.9) |
| Don't know | 131 (23.8) |
| Missing | 13 (2.4) |
| Asthma, n (%) | |
| Yes | 63 (11.4) |
| No | 459 (83.3) |
| Don't know | 23 (4.2) |
| Missing | 6 (1.1) |
| Atopic eczema, n (%) | |
| Yes | 73 (13.2) |
| No | 395 (71.7) |
| Don't know | 79 (14.3) |
| Missing | 4 (0.7) |
| Allergic rhinitis, n (%) | |
| Yes | 54 (9.8) |
| No | 441 (80.0) |
| Don't know | 49 (8.9) |
| Missing | 7 (1.3) |
| Headache, yes [a], n (%) | 185 (33.6) |
| Missing | 0 |
| Abdominal pain, yes [b], n (%) | 85 (15.4) |
| Missing | 1 (0.2) |
| Psychological distress [c], median (min-max) | 1.3 (1–3.5) |
| Psychological distress [c] (cat.), n (%) | |
| > 1.85 | 80 (14.5) |
| ≤ 1.85 | 466 (84.6) |
| Missing | 5 (0.9) |
| Accumulated number of health complaints [d], median (min-max) | 1 (0–5) |
| Missing, n (%) | 19 (3.4) |

[a] Headache minimum 1–6 days a month the last year

[b] Abdominal pain minimum once a week the last two months

[c] Psychological distress, Hopkins Symptom Checklist-10 (HSCL-10), continuous (1–4), categorized (cut-off > 1.85 indicates the presence of psychological distress)

[d] This variable includes asthma, atopic eczema, allergic rhinitis, headache, abdominal pain, psychological distress and other reported health complaints (prevalence rates): diabetes type 1 (0.2%), ADHD (0.6%), psoriasis (0.4%), arthritis (0.2%), anaemia (0.4%), sleep disorder (0.2%), food allergy or intolerance (1.9%)

**Table 2. Logistic regression analyses of the association between health complaints at baseline and persistent musculoskeletal pain [a] at 2-year follow-up.**

| Health complaints | Exposed cases [f] | Crude | P-value | Adjusted [g] | P-value |
|---|---|---|---|---|---|
| | | OR (95% CI) | | OR (95% CI) | |
| Asthma [b] | 12 | 1.56 (0.79, 3.10) | 0.20 | 1.71 (0.85, 3.43) | 0.13 |
| Atopic eczema [b] | 12 | 1.26 (0.64, 2.47) | 0.50 | 1.09 (0.53, 2.21) | 0.82 |
| Allergic rhinitis [b] | 10 | 1.49 (0.71, 3.10) | 0.29 | 1.34 (0.63, 2.83) | 0.45 |
| Headache [c] | 31 | 1.44 (0.87, 2.36) | 0.15 | 1.23 (0.72, 2.10) | 0.45 |
| Abdominal pain [d] | 20 | 2.29 (1.29, 4.08) | 0.01 | 2.33 (1.29, 4.19) | 0.01 |
| Psychological distress (1–4) [e] | - | 1.25 (0.75, 2.09) | 0.39 | 1.03 (0.60, 1.79) | 0.91 |

Abbreviations: Odds ratio (OR), Confidence interval (CI). N = 529 in adjusted models

[a] Persistent musculoskeletal pain is defined as weekly musculoskeletal pain for three months or more

[b] The reference is "no" or "don't know"

[c] Headache, defined as at least 1–6 days a month the last year, with less than 1–6 days a month as the reference

[d] Abdominal pain, defined as at least weekly pain the last two months, with less than once a week as the reference

[e] Psychological distress measured by the Hopkins Symptom Checklist– 10 (HSCL-10), used as a continuous variable (1–4).

[f] Number of participants with the specific health complaint at baseline *and* persistent musculoskeletal pain at follow-up

[g] Adjusted for sex, parents' employment status and other health complaints (yes/no)

Abdominal pain was associated with future persistent musculoskeletal pain, which is in accordance with a systematic review on children, adolescents and young adults, reporting abdominal pain to be a potential risk factor for future back pain [15]. In previous research, an association between headache and future musculoskeletal pain has also been found across several studies [15–18]. In our study, adolescents with headache had a slightly increased odds of future persistent musculoskeletal pain, but the association was not statistically significant. A limited number of exposed cases might explain the non-significant result. Also, the response categories for headache were different than those for abdominal pain. Headache was classified as at least one to six episodes a month, while abdominal pain was classified as at least weekly pain. Hence, the frequency of pain episodes might be important in prediction of future musculoskeletal pain. Unfortunately, a cut-off with more frequent headache was not possible in our study due to few cases.

A previous systematic review reported that negative emotional symptoms might be associated with future musculoskeletal pain in children and adolescents [14]. No statistically significant association between psychological distress and future persistent musculoskeletal pain was found in our study. However, differences in populations, exposure measurement, outcome definitions and follow-up periods make direct comparison difficult. Our results revealed no

**Table 3. Logistic regression analyses of the association between the number of health complaints at baseline and persistent musculoskeletal pain [a] at 2-year follow-up.**

| | Crude | P-value | Adjusted [c] | P-value |
|---|---|---|---|---|
| | OR (95% CI) | | OR (95% CI) | |
| Accumulated number of health complaints [b] | 1.33 (1.07, 1.65) | 0.01 | 1.33 (1.07, 1.66) | 0.01 |

Abbreviations: Odds ratio (OR), Confidence interval (CI). N = 529 in adjusted model

[a] Persistent musculoskeletal pain is defined as weekly musculoskeletal pain for three months or more

[b] Continuous variable including all reported health complaints at baseline (asthma, allergic rhinitis, atopic eczema, abdominal pain, headache, psychological distress, diabetes type 1, ADHD, psoriasis, arthritis, anaemia, sleep disorder, food allergy or intolerance)

[c] Analysis adjusted for sex and parents' employment status

statistically significant associations between allergic rhinitis or atopic eczema with future persistent musculoskeletal pain. Importantly, the number of exposed cases were small, so the results should be interpreted with caution. Nevertheless, the findings are in accordance with a previous study on 12-22-year-olds, investigating risk factors for low back pain after eight years [21]. The same study reported an association between asthma and low back pain eight years later [21]. Another study demonstrated that young adults with asthma had a non-significant increased odds for developing neck pain during a one-year follow-up [33], which is in line with our results. A potential lack of statistical power due to few exposed cases may explain the wide CI and the non-significant result in our data (OR 1.71, 95% CI [0.85–3.43]). It is not known whether different outcomes also influenced the results, as a statistically significant association was found in the study of low back pain [21], but not in our study of musculoskeletal pain in general.

We found that an accumulation of health complaints at baseline was associated with persistent musculoskeletal pain at the 2-year follow-up. To the best of our knowledge, this is the first cohort study to investigate the association between an accumulation of health complaints, including both medical diseases, psychological and somatic symptoms, and the onset of persistent musculoskeletal pain in adolescents. A few existing cohort studies have findings in accordance with our study. One former study of 12-22-year-olds found that a combination of persistent low back pain, persistent headache and asthma was strongly associated with persistent low back pain eight years later [21]. Another study of 9–12-year-olds found that for each additional physical and psychological symptom, including headache, abdominal pain, depressive mood, daytime tiredness, difficulty falling asleep and waking up during the night, the odds of weekly neck pain during a four-year follow-up increased [20]. Furthermore, previous research on adolescents have suggested that an accumulation of multiple factors in general predicts future outcomes of musculoskeletal pain more strongly than individual factors [16, 34].

The mechanisms behind the association between an accumulation of health complaints and onset of persistent musculoskeletal pain is not known. One theory is that repeated efforts to adapt to stressors over time (e.g. multiple health complaints) results in a cumulative physiological wear and tear, or allostatic overload, that disrupts physiological regulatory systems, which subsequently increase the risk of long-term health problems [19, 35], such as persistent musculoskeletal pain. Another theory might be that some health complaints share genetic mechanisms [36] or have other shared risk factors, such as childhood difficulties [37], making some individuals more vulnerable to a multitude of health complaints.

## Strengths and limitations

A strength of the study is the cohort study design, in which allows a possible understanding of the temporal sequence between exposures and outcome. Furthermore, the results are based on data from a population-based sample with a high baseline response rate, in a period of life important for the onset of persistent musculoskeletal pain.

A limitation of our study was the self-reporting of health complaints, which may have caused misclassifications. For example, misclassification of "doctor-diagnosed" medical conditions might occur if adolescents do not recall whether they have been diagnosed with the condition. Furthermore, some of the health complaints were defined with umbrella terms. For example, we did not distinguish between migraine and other types of headaches, or between more and less frequent episodes of headache (cut-off used was $\geq$ 1–6 days a month). Furthermore, we did not have information on the severity of asthma, allergic rhinitis and atopic eczema. Thus, we do not know whether distinguishing between more specific complaints or only including adolescents with the most severe symptoms could have changed the results.

The variable reflecting the number of health complaints was created by us on information of the different health complaints and is not formally validated. Another limitation of the study is low statistical power in the analyses of specific health complaints due to few exposed cases.

We adjusted for parents'employment status in the analyses as an indicator of socioeconomic status, despite some limitations. First, due to few cases in each category for those who were not employed (unemployed, disabled, working at home (housewife), attends school/ courses, retired, deceased and other), all categories were merged and made up one category. However, for example unemployment and being a "housewife" might have different reasons and indicate different levels of resources. Second, 3.6% of the adolescents did not know their parents' employment status. However, employment status was our best alternative as an indicator of socioeconomic status in the data. Almost one third of the participants were lost to follow-up or had incomplete outcome data. The group lost to follow-up or with incomplete outcome data had a lower proportion of girls, a lower proportion of adolescents with at least one parent with higher education, and a higher proportion who did not know the educational level of their parents. However, there were no statistically significant differences in prevalence of specific health complaints or in the number of health complaints between this group and the study sample. Persistent musculoskeletal pain was measured by a questionnaire developed for the Fit Futures study and is not formally validated.

## Implications

Having abdominal pain was associated with persistent musculoskeletal pain onset in adolescents. Also, an accumulation of health complaints was associated with onset of persistent musculoskeletal pain. Health care providers consulting adolescents might need to take preventive actions in those with abdominal pain and/or an accumulation of health complaints, irrespective of type, to avoid development of persistent musculoskeletal pain.

This study contributes to knowledge of the relationship between health complaints and development of persistent musculoskeletal pain in adolescence. The potential multimorbidity perspective of adolescent musculoskeletal pain is an important topic for future research to understand the underlying patterns of persistent pain conditions in youth. However, our study should be replicated by others. To enable development of appropriate preventive strategies, exploration of the mechanisms behind the associations are other potential topics for future research.

## Conclusions

Having abdominal pain in the first year of upper-secondary school was associated with persistent musculoskeletal pain 2 years later. Also, an accumulation of health complaints was associated with persistent musculoskeletal pain after 2 years. Health care providers might need to take preventive actions in adolescents with abdominal pain, and in adolescents with an accumulation of health complaints, to prevent development of persistent musculoskeletal pain. The potential multimorbidity perspective of adolescent musculoskeletal pain is an important topic for future research.

## Supporting information

**S1 Table. Comparison of baseline characteristics of respondents and non-respondents of the follow-up study (FF2).**
(DOCX)

**S2 Table. Logistic regression analyses of the associations between health complaints (asthma, atopic eczema, allergic rhinitis) at baseline and persistent musculoskeletal pain at follow-up, excluding participants who responded "don't know" to the baseline health complaints questions.**
(DOCX)

## Acknowledgments

We thank all the participants in the Fit Futures Study, the research technicians at the Clinical Research Unit at the University Hospital of North Norway for conducting the data collection and the Fit Futures administration.

## Author Contributions

**Conceptualization:** Kaja Smedbråten, Margreth Grotle, Henriette Jahre, Kåre Rønn Richardsen, Pierre Côté, Ólöf Anna Steingrímsdóttir, Kjersti Storheim, Christopher Sivert Nielsen, Britt Elin Øiestad.

**Formal analysis:** Kaja Smedbråten.

**Funding acquisition:** Margreth Grotle, Britt Elin Øiestad.

**Investigation:** Christopher Sivert Nielsen.

**Methodology:** Kaja Smedbråten, Margreth Grotle, Henriette Jahre, Kåre Rønn Richardsen, Pierre Côté, Ólöf Anna Steingrímsdóttir, Kjersti Storheim, Christopher Sivert Nielsen, Britt Elin Øiestad.

**Project administration:** Britt Elin Øiestad.

**Supervision:** Margreth Grotle, Britt Elin Øiestad.

**Visualization:** Kaja Smedbråten, Margreth Grotle, Henriette Jahre, Kåre Rønn Richardsen, Pierre Côté, Ólöf Anna Steingrímsdóttir, Kjersti Storheim, Christopher Sivert Nielsen, Britt Elin Øiestad.

**Writing – original draft:** Kaja Smedbråten.

**Writing – review & editing:** Kaja Smedbråten, Margreth Grotle, Henriette Jahre, Kåre Rønn Richardsen, Pierre Côté, Ólöf Anna Steingrímsdóttir, Kjersti Storheim, Christopher Sivert Nielsen, Britt Elin Øiestad.

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
