## [Decision Letter · Decision Letter 0]

13 Jul 2022

PONE-D-22-17250Adolescents with health complaints are more likely to develop persistent musculoskeletal pain: The Fit Futures StudyPLOS ONE

Dear Dr. Smedbråten,

Thank you for submitting your manuscript to PLOS ONE. After careful consideration, we feel that it has merit but does not fully meet PLOS ONE’s publication criteria as it currently stands. Therefore, we invite you to submit a revised version of the manuscript that addresses the points raised during the review process.

We look forward to receiving your revised manuscript.

Kind regards,

Dong Keon Yon, MD, FACAAI

Academic Editor

PLOS ONE

Journal Requirements:

Additional Editor Comments:

Thank you for submitting your manuscript. The reviewers and I believe it is of potential value for our readers. However, the reviewers have raised a number of very important issues, and their excellent comments will need to be adequately addressed in a revision before the acceptability of your manuscript for publication in the Journal can be determined. We cannot guarantee that your revised paper will be chosen for publication; this would be solely based on how satisfactorily you have addressed the reviewer comments.

#1. The evidence level of no association is low due to the small event numbers of this study (i.e., Total=500 / Event=10~15, these sample may lead to small sample bias).

So please tone down the main results.

I.e.,

Our analyses showed no significant associations between asthma, allergic rhinitis, eczema, headache or psychological distress and persistent musculoskeletal pain onset.

-> Our analyses showed low evidence of potential associations between asthma, allergic rhinitis, eczema, headache or psychological distress and persistent musculoskeletal pain onset, although the event numbers were small.

#2. The authors have to cite the statistical guideline such as https://doi.org/10.54724/lc.2022.e3

Thank you.

Reviewers' comments:

Reviewer's Responses to Questions

**Comments to the Author**

1. Is the manuscript technically sound, and do the data support the conclusions?

Reviewer #1: Partly

Reviewer #2: Yes

2. Has the statistical analysis been performed appropriately and rigorously? 

Reviewer #1: Yes

Reviewer #2: Yes

3. Have the authors made all data underlying the findings in their manuscript fully available?

Reviewer #1: No

Reviewer #2: Yes

4. Is the manuscript presented in an intelligible fashion and written in standard English?

Reviewer #1: Yes

Reviewer #2: Yes

5. Review Comments to the Author

Reviewer #1: This study aims to assess whether specific health complaints and an accumulation of 33 health complaints were associated with the onset of persistent musculoskeletal pain 2 years later.

The work is systematically well done by analyzing the effect of each health complaints on musculoskeletal pain.

My main concern is the necessity of this research. Although it is interesting to note that Adolescents with health complaints are more likely to 2 develop persistent musculoskeletal pain, I still wonder what this research will lead us to.

I am also wondering if could self-reported questionnaire, “Do you have persistent or constantly recurring pain that has lasted for 3 months or more?” or “ Has a doctor ever told you that you have...”, work as a proper criterion. It would be great if the authors can provide some references to questions themselves to support the background.

P values should follow the reporting of comparisons of absolute numbers or rates and measures of uncertainty.

There is a lack of explanation for the reasons for the health complaints that have been linked to musculoskeletal pain. It would be nice to supplement it through an intensive literature review.

Reviewer #2: This paper is interesting and technically well performed. I think it needs only few corrections.

Abstract

Both in the abstract and in the text, it should be stated that health complaints such as asthma, allergic rhinitis, eczema, headache, abdominal pain, psychological distress, AND OTHERS are assessed. Otherwise, it seems that only 5 possible health complaints are assessed.

Materials and Methods

108-109: Moreover, 202 participants reported persistent musculoskeletal pain at baseline and were not at risk of developing the outcome of interest.

133-134: The measurement of persistent musculoskeletal pain at baseline was only used to identify the population at risk in our cohort study.

I think that saying “the population at risk” is not correct. The population is at risk of starting chronic musculoskeletal pain but those who are excluded already have chronic musculoskeletal pain. It is more understandable that the 202 patients with baseline pain are excluded from the sample and that the measurement of persistent pain was only used to define the study population, without pain at baseline.

173-176: The accumulated number of health complaints was analysed as a continuous variable and included the specific health complaints investigated in this study (asthma, allergic rhinitis, eczema, psychological distress, headache, and abdominal pain) and other complaints captured through “other health complaints”.

“Other health complaints” were collected as a yes or no variable. In the case of having more than one health complaint (apart from the 5 specific ones), was the number collected for analysis? If it did not occur in any case, indicate it in the results.

Results

245: The two-year incidence of persistent musculoskeletal pain was 13.8 % (95% CI 11.1, 17.0).

Adding the number of cases, not just the percentage, is not in the table either.

Discussion

283-286: An association between psychological distress and future musculoskeletal pain has been demonstrated in previous studies [13], but was not found in our study. Differences in populations, exposure measurement, outcome definitions and follow-up periods might explain the discrepancy in results.

The results may be influenced by the fact that patients who already have pain at baseline have been excluded from the study (202). This group probably has a higher percentage of health complaints, including psychological distress, and its inclusion would give more statistical power to the study.

It would be interesting to have the data of this group of patients with pain at the beginning, how many were still in pain at the end of follow-up and how many had other health complaints.

300-303: To the best of our knowledge, this is the first cohort study to investigate the association between an accumulation of health complaints and persistent musculoskeletal pain in adolescents, including both medical diagnoses, psychological and somatic symptoms.

You have to write … the association between an accumulation of health complaints and THE ONSET OF persistent musculoskeletal pain.

6. PLOS authors have the option to publish the peer review history of their article (what does this mean?). If published, this will include your full peer review and any attached files.

Reviewer #1: **Yes: **Gun Ahn

Reviewer #2: No

---

## [Author Response · Author response to Decision Letter 0]

13 Nov 2022

Author response: We have ensured that our manuscript meets PLOS ONE’s style requirements, including those for file naming. 

Author action: Our manuscript is changed to meet PLOS ONE’s style requirements.

Author response: We have added information on the type of the participant consent in the ethics statement in the Methods and online submission information. Information on consent from parents or guardians was already described in the paper.

Author action: 

Page 4, line 127 – 128: We have added information on the type of the participant consent: “All participants provided written informed consent prior to participation. For participants under the age of 16 years, written consent of one parent/guardian was also obtained.”

Editor comments

1. The evidence level of no association is low due to the small event numbers of this study (i.e., Total=500 / Event=10~15, these sample may lead to small sample bias).

So please tone down the main results. Our analyses showed no significant associations between asthma, allergic rhinitis, eczema, headache or psychological distress and persistent musculoskeletal pain onset.

-> Our analyses showed low evidence of potential associations between asthma, allergic rhinitis, eczema, headache or psychological distress and persistent musculoskeletal pain onset, although the event numbers were small.

Author response: Thank you for an important consideration. We agree on toning down the main results by including the following: 1) When referring to the associations which were not statistically significant, we describe that these were “not statistically significantly associated” instead of describing them as “not associated”. 2) We have toned down the non-significant results in the discussion and emphasised the low number of exposed cases as potential reasons for non-significant results. 3) The topic is emphasised in the limitation section.

Author action: 

1) We have emphasised that the results which were not associated were not statistically significantly associated:

Page 1, line 48 – 50: “Our analyses showed no statistically significant associations between asthma, allergic rhinitis, atopic eczema, headache or psychological distress and persistent musculoskeletal pain at the 2-year follow-up.”

2) We have toned down the non-significant results in the discussion and emphasised the low number of exposed cases:

Page 12, line 290 – 294: “In previous research, an association between headache and future musculoskeletal pain has also been found across several studies [1-4]. In our study, adolescents with headache had a slightly increased odds of future persistent musculoskeletal pain, but the association was not statistically significant. A limited number of exposed cases might explain the non-significant result.”

Page 13, line 305 – 308: “Our results revealed no statistically significant associations between allergic rhinitis or atopic eczema with future persistent musculoskeletal pain. Importantly, the number of exposed cases were small, so the results should be interpreted with caution.”

Page 13, line 310 – 317: 

“The same study reported an association between asthma and low back pain eight years later [5]. Another study demonstrated that young adults with asthma had a non-significant increased odds for neck pain at one-year follow-up [6], which is in line with our results. A potential lack of statistical power due to few exposed cases may explain the wide CI and the non-significant result in our data (OR 1.70, 95% CI [0.85 – 3.42]). It is not known whether different outcomes also influenced the results, as a statistically significant association was found in the study of low back pain [5], but not in our study of musculoskeletal pain in general.”

3) The topic is emphasized in the limitation section:

Page 15, line 345 – 347: “Another limitation of the study is low statistical power in the analyses of specific health complaints due to few exposed cases, particularly for asthma, allergic rhinitis and eczema.”

2. The authors have to cite the statistical guideline such as https://doi.org/10.54724/lc.2022.e3

Author response: Thank you for this suggestion. We have now added a reference to a statistical guideline.

Author action: 

Page 8, line 222: A reference to a statistical guideline has been added: “Logistic regression analyses [7] were used to measure the association between baseline health complaints and the presence of persistent musculoskeletal pain at the 2-year follow-up.”

Reviewer #1: 

1) My main concern is the necessity of this research. Although it is interesting to note that Adolescents with health complaints are more likely to 2 develop persistent musculoskeletal pain, I still wonder what this research will lead us to.

Author response: Thank you for this important comment. We will try to make it clearer why we consider this research as an important contribution to the field of musculoskeletal pain. Research on persistent musculoskeletal pain in adolescents is sparse and in contrast to the high prevalence of pain conditions and pain killer usage among younger people. We believe the results of this study add to the understanding of the overall picture of health in adolescence by investigating the associations between different prevalent health complaints and development of musculoskeletal pain. Recognized papers, such as the Lancet – series on low back pain, have emphasised that adults with other health complaints are more likely to report back pain than people in good health, but that the mechanisms behind the coexistence are not known [8], and a systematic review has asked for increased knowledge regarding this topic in youth [1]. We consider this study as a first step to contemplate persistent musculoskeletal pain in adolescence in a multimorbidity perspective. The next research step, with this study’s findings in mind, would be to investigate mechanisms of this connection to be able to find potential targets for preventive interventions. This is of particular importance, as we know multimorbidity in general is associated with poor outcomes and health care utilization later in life [9-11].

In a clinical perspective, we consider the findings in this paper important as it tells us that a broad assessment of health might be important in examination of adolescents. Adolescents with co-occurrent health complaints might be a subgroup with increased vulnerability of developing persistent pain, in which is important for future development of preventive strategies. 

We have added a sentence in the conclusion of the abstract and in the main conclusion of the paper to make our argument of the importance of the knowledge from this paper clearer to the reader.

Author action: 

Page 1, line 53 – 57: We have added information to the conclusion in the abstract: 

“Health care providers might need to take preventive actions in adolescents with abdominal pain and in adolescents with an accumulation of health complaints, to prevent development of persistent musculoskeletal pain. The potential multimorbidity perspective of adolescent musculoskeletal pain is an important topic for future research to understand the underlying patterns of persistent pain conditions in adolescents.”

Page 17, line 399 – 400: We have added a sentence in the main conclusion:

“The potential multimorbidity perspective of adolescent musculoskeletal pain is an important topic for future research.”

2) I am also wondering if could self-reported questionnaire, “Do you have persistent or constantly recurring pain that has lasted for 3 months or more?” or “ Has a doctor ever told you that you have...”, work as a proper criterion. It would be great if the authors can provide some references to questions themselves to support the background.

Author response: Thank you for an important question. The question assessing persistent musculoskeletal pain; “Do you have persistent or recurring pain that has lasted for 3 months or more?”, is similar to the description of chronic pain in the ICD-11, defined as “persistent or recurrent pain lasting longer than 3 months” [12]. This specific question was, however, developed for the Fit Futures study and, unfortunately, it is not formally validated. The potential limitation of using an unvalidated outcome measure is emphasised in the limitation section: 

Page 16, line 377 – 379: “Persistent musculoskeletal pain was measured by a questionnaire developed for the Fit Futures study and is not formally validated.”

Concerning the question assessing asthma, allergy and eczema; “ Has a doctor ever told you that you have...” is a well-used question on self-reported health complaints in observational health studies. For example, in the adolescent part of “the Trøndelag Health study (HUNT)”, an international well-recognised health study, similar questions have been used to measure self-reported health complaints [13]. Unfortunately, the question is not formally validated in adolescents. The potential limitation of self-reported health complaints is emphasised in the limitation section: 

Page 14, line 349 – 351: “A limitation of our study was the self-reporting of health complaints, which may have caused misclassifications. For example, misclassification of “doctor-diagnosed” medical conditions might occur if adolescents do not recall whether they have been diagnosed with the condition.”

3) P values should follow the reporting of comparisons of absolute numbers or rates and measures of uncertainty.

Author response: We have added p-values in tables and text.

Author action: 

We have added p-values in the abstract:

Page 1, line 47 - 48: “Baseline abdominal pain was associated with persistent musculoskeletal pain 2 years later (OR 2.31, 95% CI [1.29-4.17], p=0.01). 

Page 1, line 50 – 53: “An accumulation of health complaints at baseline increased the odds for persistent musculoskeletal pain by 33% for each additional health complaint (OR 1.33, 95% CI [1.07-1.66], p=0.01).”

Page 11-12: We have added p-values in Table 1 and Table 2

We have added p-values in the text in the results chapter, following the OR-estimates: 

Page 10, line 260 – 262: “Participants with baseline abdominal pain had 2.31 (95% CI [1.29 - 4.17], p = 0.01) times the odds of persistent musculoskeletal pain at follow-up compared to those without abdominal pain “

Page 11 - 12, line 276 – 278: “Specifically, for each additional reported complaint at baseline, the odds of reporting persistent musculoskeletal pain at the 2-year follow-up increased with 33% (OR 1.33, 95% CI [1.07 - 1.66], p=0.01).”

4) There is a lack of explanation for the reasons for the health complaints that have been linked to musculoskeletal pain. It would be nice to supplement it through an intensive literature review.

Author response: Thank you for a well-thought suggestion. Just to clarify; all health complaints reported by the adolescents are investigated against musculoskeletal pain in this study through the accumulated number of health complaints, but only the most common health complaints were also investigated as separate exposures. 1) To emphasise that all present health complaints are included in the variable of accumulated health complaints, we have rewritten the text regarding this variable. 2) To give a better explanation of why we have examined the six specific health complaints, we have now added information in the introduction to argue that these are six common health complaints in adolescence. 3) Literature on the association between the specific health complaints we have chosen to look specifically into, and musculoskeletal pain, is presented in the introduction, based on results from existing systematic reviews.

Author action:

1) Page 7, line 187-190: The text regarding the variable on the accumulated number of health complaints has been rewritten to better emphasise that all present health complaints are included:

“The accumulated number of health complaints was analysed as a continuous variable and included any health complaints reported at baseline (asthma, allergic rhinitis, atopic eczema, psychological distress, headache, abdominal pain and any other health complaints captured through the question “do you have any chronic or persistent disease?”).”

2) Page 2, line 71 – 76: We have added a section to the introduction to emphasise that the health complaints we assess individually in this study are common health complaints among adolescents: 

“Importantly, also headache [14, 15], abdominal pain [15] and mental health problems [16] are all common health complaints in children and adolescents. Furthermore, medical conditions, such as asthma, eczema [17, 18] and allergic rhinitis [18] are frequent. Several of these health complaints commonly co-occur with musculoskeletal pain [19-21], however, whether the presence of different health complaints increase the risk of developing musculoskeletal pain is unclear.”

3) Page 2, line 76 - 81: Literature on the association between specific health complaints and musculoskeletal pain in adolescents is presented in the introduction:

“A systematic review suggested that negative emotional symptoms might increase the risk of developing musculoskeletal pain in children and adolescents [22]. Studies have also indicated an association between abdominal pain [1] and headache [1-4] with later musculoskeletal pain. For the association between other health issues, including medical conditions, and musculoskeletal pain onset in children and adolescents, the existing findings are inconsistent [22].”

Reviewer #2: 

1) Abstract

Both in the abstract and in the text, it should be stated that health complaints such as asthma, allergic rhinitis, eczema, headache, abdominal pain, psychological distress, AND OTHERS are assessed. Otherwise, it seems that only 5 possible health complaints are assessed.

Author response: Thank you for making us aware that this was not clearly defined. We have rewritten the text in the abstract to make this clearer. We have also tried to make the definition of “the accumulated number of health complaints” clearer in other parts of the article.

Author action:

We have added “other reported health complaints” to the text to make it clearer that other health complaints than the 6 specific complaints are included in the “accumulated number of health complaints” – variable: 

Page 1, line 40 - 45 (in the abstract): 

“The following self-reported baseline health complaints were investigated as individual exposures: asthma, allergic rhinitis, atopic eczema, headache, abdominal pain and psychological distress. We also investigated the association between an accumulated number of these and other reported health complaints, and persistent musculoskeletal pain 2 years later.”

Page 8, line 224 – 227 (in the method section): 

“We analysed crude and adjusted associations with persistent musculoskeletal pain in separate models for specific health complaints (asthma, allergic rhinitis, atopic eczema, psychological distress, headache and abdominal pain) and the accumulated number of these and other reported baseline health complaints.”

Page 7, line 187 – 190: We have rewritten the definition of “the accumulated number of health complaints”: 

“The accumulated number of health complaints was analysed as a continuous variable and included any health complaints reported at baseline (asthma, allergic rhinitis, eczema, psychological distress, headache, abdominal pain and any other health complaints captured through the question “do you have any chronic or persistent disease?”).”

2) Materials and Methods

108-109: Moreover, 202 participants reported persistent musculoskeletal pain at baseline and were not at risk of developing the outcome of interest.

133-134: The measurement of persistent musculoskeletal pain at baseline was only used to identify the population at risk in our cohort study.

I think that saying “the population at risk” is not correct. The population is at risk of starting chronic musculoskeletal pain but those who are excluded already have chronic musculoskeletal pain. It is more understandable that the 202 patients with baseline pain are excluded from the sample and that the measurement of persistent pain was only used to define the study population, without pain at baseline.

Author response: Thank you for this comment. We have tried to rewrite to make this clearer.

Author action: 

Page 4, line 117 - 122: We have rewritten the text: 

“Of 1117 adolescents invited to participate in FF1, 1038 (92.9%) responded. Thirty-six participants were excluded from our sample because they were older than 19 years of age at baseline. Moreover, 190 participants reported persistent musculoskeletal pain at baseline and were excluded from the sample. Finally, 23 participants were excluded because they had incomplete data on musculoskeletal pain at baseline (Fig 1). Of 789 respondents to the FF1 Study that met the inclusion criteria, 549 (69.6 %) participated in FF2 and had complete outcome data for the present study.”

Page 5, line 144 – 146: We have rewritten the text:

“Persistent musculoskeletal pain was also measured and defined as described above at baseline. The measurement of persistent musculoskeletal pain at baseline was only used to identify the sample in our cohort study.”

Page 8, line 218 – 220: We have rewritten the text:

“Incidence rate with 95% confidence intervals (CIs) of persistent musculoskeletal pain was computed as the number of participants who reported persistent musculoskeletal pain at the 2-year follow-up divided by the total number of the study sample and reported as percentage.”

3) 173-176: The accumulated number of health complaints was analysed as a continuous variable and included the specific health complaints investigated in this study (asthma, allergic rhinitis, eczema, psychological distress, headache, and abdominal pain) and other complaints captured through “other health complaints”.

“Other health complaints” were collected as a yes or no variable. In the case of having more than one health complaint (apart from the 5 specific ones), was the number collected for analysis? If it did not occur in any case, indicate it in the results.

Author response: Thank you for making us aware that this was not clearly defined in the text. “Other health complaints” were not collected as a yes or no variable. The question “Do you have any chronic or persistent disease?” was asked, and all reported diseases / complaints were recoded to ICD-codes and reported separately. The variable of the accumulated number of health complaints includes the precise number of any reported health complaints at baseline. We have tried to rewrite the text to make this clearer. We have also changed Table 1 by removing the variable called “other health complaints”, as this variable description might confuse the reader to think other health complaints have only been treated as a yes/no variable. 

Author action: We have rewritten the text:

Page 6 - 7, line 180 – 185: 

“Other health complaints were measured by the question “Do you have any chronic or persistent disease?” The question was asked during an interview by a nurse and responses were recoded to ICD-10 codes. The prevalence of the other health complaints reported through this question, were too low to analyse as individual exposures, but they were included as separate health complaints when investigating the accumulated number of health complaints at baseline.”

Page 6-7, line 187 – 190: 

“The accumulated number of health complaints was analysed as a continuous variable and included any health complaints reported at baseline (asthma, allergic rhinitis, eczema, psychological distress, headache, abdominal pain and any other health complaints captured through the question “do you have any chronic or persistent disease?”)”

Page 9 - 10, Table 1: The variable “other health complaints yes/no” was removed from Table 1. We have rather included a footnote on which other health complaints that have been reported and linked this footnote to the “accumulated number of health complaints” variable:

“f The variable includes asthma, atopic eczema, allergic rhinitis, headache, abdominal pain, psychological distress and other reported health complaints (prevalence rates): diabetes type 1 (0.2%), ADHD (0.6%), psoriasis (0.4%), arthritis (0.2%), anemia (0.4%), sleep disorder (0.2%), food allergy or intolerance (1.9%)”

4) Results

245: The two-year incidence of persistent musculoskeletal pain was 13.8 % (95% CI 11.1, 17.0). Adding the number of cases, not just the percentage, is not in the table either.

Author response: Thank you for this comment. The number of cases is now included.

Author action:

Page 10, line 259 – 260: We have rewritten the text to add information on the number of cases: “The two-year incidence of persistent musculoskeletal pain was 13.8 % (95% CI [11.2- 17.0]), (n=76).”

5) Discussion

283-286: An association between psychological distress and future musculoskeletal pain has been demonstrated in previous studies [13], but was not found in our study. Differences in populations, exposure measurement, outcome definitions and follow-up periods might explain the discrepancy in results.

The results may be influenced by the fact that patients who already have pain at baseline have been excluded from the study (202). This group probably has a higher percentage of health complaints, including psychological distress, and its inclusion would give more statistical power to the study. It would be interesting to have the data of this group of patients with pain at the beginning, how many were still in pain at the end of follow-up and how many had other health complaints.

Author response:

We agree that this would be interesting. However, to be able to say something about the temporal sequence between exposure and outcome, we find it important to exclude the group of individuals with baseline musculoskeletal pain in our analyses. In systematic reviews of risk factors, this is often a criterium for inclusion [2, 22, 23].

Regarding the association between psychological distress and persistent musculoskeletal pain, the results might indeed be different for studies including those with baseline pain or not. On the other hand, a systematic review, in which only studies on participants without baseline pain were included, concluded that emotional symptoms might be a risk factor for musculoskeletal pain in children and adolescents [22]. Hence, excluding participants with baseline pain cannot be the only reason why we do not find any association in our data.

If we understand the comment right, the reviewer also found it interesting to include the participants with baseline musculoskeletal pain for descriptive purposes. We agree that this would be interesting. However, as we have defined our population as those without persistent musculoskeletal pain at baseline, we find it a bit complicated to include this group for descriptive purposes in a clear way for the reader. Hence, we prefer not to include these participants to prevent misunderstandings on which sample we have analysed. Please let us know if these additional descriptive analyses are of particularly importance to present.

6) 300-303: To the best of our knowledge, this is the first cohort study to investigate the association between an accumulation of health complaints and persistent musculoskeletal pain in adolescents, including both medical diagnoses, psychological and somatic symptoms.

You have to write … the association between an accumulation of health complaints and THE ONSET OF persistent musculoskeletal pain.

Author response: We have now added “the onset of” in the text.

Author action:

Page 14, line 320 – 323: We have added “the onset of” in the text:

“To the best of our knowledge, this is the first cohort study to investigate the association between an accumulation of health complaints, including both medical diseases, psychological and somatic symptoms, and the onset of persistent musculoskeletal pain”

1. Beynon AM, Hebert JJ, Hodgetts CJ, Boulos LM, Walker BF. Chronic physical illnesses, mental health disorders, and psychological features as potential risk factors for back pain from childhood to young adulthood: a systematic review with meta-analysis. European Spine Journal. 2020;29(3):480-96 doi:10.1007/s00586-019-06278-6

2. Jahre HG, M. Småstuen, M. Guddal, M. H. Smedbråten, K. Richardsen, K. R. Stensland, S. Storheim, K. Øiestad, B. E. Risk factors and risk profiles for neck pain in young adults: Prospective analyses from adolescence to young adulthood—The North-Trøndelag Health Study. PLOS One. 2021;16(8):e0256006 doi:10.1371/journal.pone.0256006

3. Jones GT, Silman AJ, Macfarlane GJ. Predicting the onset of widespread body pain among children. Arthritis & Rheumatism. 2003;48(9):2615-21 doi:10.1002/art.11221

4. El-Metwally A, Salminen JJ, Auvinen A, Macfarlane G, Mikkelsson M. Risk factors for development of non-specific musculoskeletal pain in preteens and early adolescents: a prospective 1-year follow-up study. BMC Musculoskeletal Disorders. 2007;8:46 doi:10.1186/1471-2474-8-46

5. Hestbaek L, Leboeuf-Yde C, Kyvik KO. Is comorbidity in adolescence a predictor for adult low back pain? A prospective study of a young population. BMC Musculoskeletal Disorders. 2006;7:29 doi:10.1186/1471-2474-7-29

6. Grimby-Ekman A, Andersson EM, Hagberg M. Analyzing musculoskeletal neck pain, measured as present pain and periods of pain, with three different regression models: a cohort study. BMC Musculoskeletal Disorders. 2009;10:73 doi:10.1186/1471-2474-10-73

7. Vittinghoff E, Glidden DV, Shiboski SC, McCulloch CE, SpringerLink. Regression methods in biostatistics : linear, logistic, survival, and repeated measures models. 2nd ed. ed. New York: Springer; 2012.

8. Hartvigsen J, Hancock MJ, Kongsted A, Louw Q, Ferreira ML, Genevay S, et al. What low back pain is and why we need to pay attention. Lancet (London, England). 2018;391(10137):2356-67 doi:10.1016/s0140-6736(18)30480-x

9. Soley-Bori M, Ashworth M, Bisquera A, Dodhia H, Lynch R, Wang Y, et al. Impact of multimorbidity on healthcare costs and utilisation: a systematic review of the UK literature. British Journal of General Practice. 2021;71(702):e39-e46 doi:10.3399/bjgp20X713897

10. Fortin M, Lapointe L, Hudon C, Vanasse A, Ntetu AL, Maltais D. Multimorbidity and quality of life in primary care: a systematic review. Health and Quality of Life Outcomes. 2004;2(1):51 doi:10.1186/1477-7525-2-51

11. Ryan A, Wallace E, O'Hara P, Smith SM. Multimorbidity and functional decline in community-dwelling adults: a systematic review. Health Qual Life Outcomes. 2015;13:168 doi:10.1186/s12955-015-0355-9

12. Treede R-D, Rief W, Barke A, Aziz Q, Bennett MI, Benoliel R, et al. A classification of chronic pain for ICD-11. Pain. 2015;156(6) 

13. Holmen T, Bratberg G, Krokstad S, Langhammer A, Hveem K, Midthjell K, et al. Cohort profile of the Young-HUNT Study, Norway: A population-based study of adolescents. International journal of epidemiology. 2013;43 doi:10.1093/ije/dys232

14. Wöber-Bingöl Ç. Epidemiology of Migraine and Headache in Children and Adolescents. Current Pain and Headache Reports. 2013;17(6):341 doi:10.1007/s11916-013-0341-z

15. King S, Chambers CT, Huguet A, MacNevin RC, McGrath PJ, Parker L, et al. The epidemiology of chronic pain in children and adolescents revisited: A systematic review. Pain. 2011;152(12):2729-38 doi:10.1016/j.pain.2011.07.016

16. Silva SA, Silva SU, Ronca DB, Gonçalves VSS, Dutra ES, Carvalho KMB. Common mental disorders prevalence in adolescents: A systematic review and meta-analyses. PlOS ONE. 2020;15(4):e0232007-e doi:10.1371/journal.pone.0232007

17. Hansen TE, Evjenth B, Holt J. Increasing prevalence of asthma, allergic rhinoconjunctivitis and eczema among schoolchildren: three surveys during the period 1985-2008. Acta paediatrica (Oslo, Norway : 1992). 2013;102(1):47-52 doi:10.1111/apa.12030

18. Hill DA, Grundmeier RW, Ram G, Spergel JM. The epidemiologic characteristics of healthcare provider-diagnosed eczema, asthma, allergic rhinitis, and food allergy in children: a retrospective cohort study. BMC Pediatr. 2016;16:133 doi:10.1186/s12887-016-0673-z

19. Jahre H, Grotle, M., Smedbråten, K., Richardsen, K.R., Bakken, A., Øiestad, B.E. Neck and shoulder pain in adolescents seldom occur alone: Results from the Norwegian Ungdata Survey. European Journal of Pain. 2021;25(8):1751-9 doi:https://doi.org/10.1002/ejp.1785

20. Swain MS, Henschke N, Kamper SJ, Gobina I, Ottová-Jordan V, Maher CG. An international survey of pain in adolescents. BMC Public Health. 2014;14(1):447 doi:10.1186/1471-2458-14-447

21. Hestbaek L, Leboeuf-Yde C, Kyvik KO, Vach W, Russell MB, Skadhauge L, et al. Comorbidity With Low Back Pain: A Cross-sectional Population-Based Survey of 12- to 22-Year-Olds. Spine. 2004;29(13):1483-91 doi:10.1097/01.BRS.0000129230.52977.86

22. Huguet A, Tougas ME, Hayden J, McGrath PJ, Stinson JN, Chambers CT. Systematic review with meta-analysis of childhood and adolescent risk and prognostic factors for musculoskeletal pain. Pain. 2016;157(12):2640-56 doi:10.1097/j.pain.0000000000000685

23. Oiestad BE, Hilde G, Tveter AT, Peat GG, Thomas MJ, Dunn KM, et al. Risk factors for episodes of back pain in emerging adults. A systematic review. European journal of pain (London, England). 2020;24(1):19-38 doi:10.1002/ejp.1474

---

## [Decision Letter · Decision Letter 1]

28 Nov 2022

Adolescents with health complaints are more likely to develop persistent musculoskeletal pain: The Fit Futures Study

PONE-D-22-17250R1

Dear Dr. Smedbråten,

We’re pleased to inform you that your manuscript has been judged scientifically suitable for publication and will be formally accepted for publication once it meets all outstanding technical requirements.

Kind regards,

Dong Keon Yon, MD, FACAAI

Academic Editor

PLOS ONE

Additional Editor Comments (optional):

This is an excellent paper.

Reviewers' comments:

Reviewer's Responses to Questions

**Comments to the Author**

1. If the authors have adequately addressed your comments raised in a previous round of review and you feel that this manuscript is now acceptable for publication, you may indicate that here to bypass the “Comments to the Author” section, enter your conflict of interest statement in the “Confidential to Editor” section, and submit your "Accept" recommendation.

Reviewer #1: All comments have been addressed

Reviewer #2: All comments have been addressed

2. Is the manuscript technically sound, and do the data support the conclusions?

Reviewer #1: Yes

Reviewer #2: Yes

3. Has the statistical analysis been performed appropriately and rigorously? 

Reviewer #1: Yes

Reviewer #2: Yes

4. Have the authors made all data underlying the findings in their manuscript fully available?

Reviewer #1: Yes

Reviewer #2: Yes

5. Is the manuscript presented in an intelligible fashion and written in standard English?

Reviewer #1: Yes

Reviewer #2: Yes

6. Review Comments to the Author

Reviewer #1: It would be interesting to do further research on why abdominal pain is associated with persistent musculoskeletal pain in adolescence

Reviewer #2: (No Response)

7. PLOS authors have the option to publish the peer review history of their article (what does this mean?). If published, this will include your full peer review and any attached files.

Reviewer #1: **Yes: **Gun Ahn

Reviewer #2: No

---

## [Editor Report · Acceptance letter]

16 Dec 2022

PONE-D-22-17250R1 

Accumulation of health complaints is associated with persistent musculoskeletal pain two years later in adolescents:
The Fit Futures Study 

Dear Dr. Smedbråten:

I'm pleased to inform you that your manuscript has been deemed suitable for publication in PLOS ONE. Congratulations! Your manuscript is now with our production department. 

Kind regards, 

on behalf of

Dr. Dong Keon Yon 

Academic Editor

PLOS ONE